# Evaluation of Mask Performances in Filtration and Comfort in Fabric Combinations

**DOI:** 10.3390/nano13030378

**Published:** 2023-01-17

**Authors:** Ji Wang, Renhai Zhao, Yintao Zhao, Xin Ning

**Affiliations:** Industrial Research Institute of Nonwovens & Technical Textiles, College of Textiles & Clothing, Shandong Center for Engineered Nonwovens, Qingdao University, Qingdao 266071, China

**Keywords:** fabric mask, fabric combination, aerosol, filtration efficiency, filtration quality factor

## Abstract

A systemic study on improving particulate pollutant filtration efficiency through the combination of conventional fabrics is presented with the objective of finding comfortable, yet effective airway mask materials and products. Fabrics, nonwovens, and their combinations made of cotton, silk, wool, and synthetic fibers are examined on their filtration efficiency for aerosol particles with diameters ranging from 0.225 μm to 3.750 μm under industry-standard testing conditions. It is found that composite fabrics can improve filtration efficiency more than just layers of the same fabric, and the filtration quality factor of some of the fabric combinations can exceed that of the standard melt-blown materials. In addition, fabric friction and charging between the combined layers also improve filtration efficiency substantially. With a broader understanding of the fabric characteristics, we may design mask products with reduced facial skin discomfort, better aesthetics, as well as the ability to alleviate the environmental impact of discarded protective masks in the extended period of controlling the transmission of pollutants and viruses, such as during the COVID-19 pandemic.

## 1. Introduction

Nearly three years have passed since the current pandemic—the global spread of the COVID-19 virus—which has caused great harm and losses to human life as well as to the global way of life and economic functions. The main transmission route of the coronavirus is through the human respiratory tract, through the droplets and aerosols in the air and upon contact [1]. Although the flow of these aerosols is largely dependent on environmental factors, such as wind, humidity, temperature, etc., they will be suspended in the air for a long time, playing a key role in the transmission of infection [2,3]. In this context, facial masks designed to protect people from bacteria and viruses are an important line of defence against respiratory infections and are essential in preventing the spread of viruses to humans. However, the current non-woven medical masks utilised in the COVID-19 pandemic are generally uncomfortable to wear, and difficult to dispose of after use. They are difficult to recycle and do not degrade at all, so their accumulation causes long-term environmental pollution. In the early stage of the COVID-19 pandemic, medical workers often wore medical protective masks continuously for more than 12 h [4], causing serious facial indentation, so their air permeability and comfort need to be improved. Therefore, there is a scientific need to test existing household fabric materials and their combinations that have better air permeability, are economical, can provide a comfortable wearing experience for the public, and can block airborne viruses in a defined usage environment. Long-term use of face masks will also cause skin irritation in some groups such as children and the elderly. To mitigate this situation, public health agencies in some countries, such as the Centers for Disease Control and Prevention (CDC) in the United States, recommend wearing surgical masks in general public places only where safe social distancing is difficult, and homemade masks made of household fabric can be used in low-traffic situations where safe distancing is possible.

Although conventional homemade fabric masks are more comfortable and stylish, their filtration efficiency towards particulates and aerosols have not been systematically studied and it appears that no standard applies to these types of consumer products. Some studies have shown that homemade fabric masks to some extent can be effective in reducing the transmission rate of COVID-19 [5,6], but they are not systematic and in no way conclusive. It is therefore highly desirable that we find more efficient fabric materials or structural designs, so we can more effectively bring scientifically effective fabric product solutions to consumers. From the perspective of product design, due to the softness of the fabric, it is easier to achieve sufficient sealing between the mask and the wearer, which improves the effectiveness of the product, provides overall consumer protection, and effectively prevents the virus aerosol from being inhaled into the human body [7]. Only with the systematic scientific understanding and smart choices of more common, yet comfortable fabrics in product design can one be more confident in the effectiveness of these ”consumer-grade” fabric masks and use them as effective means of preventing the spread of the virus, according to guidance from the Centers for Disease Control and Prevention (CDC) and the World Health Organization (WHO).

Worldwide, the use of fabrics and assemblages as aerosol filtration media has already gained interest in the past few years. Cotton fabric is the most common material for homemade masks. Some scholars have found that the filtration efficiency of cotton for NaCl aerosol with a diameter of 50~10 µm is 20~70%; the filtration efficiency of cotton for spherical latex particles with a diameter of 100 nm is, however, very low at just 10%; and the filtration efficiency of cotton for diesel soot with a diameter of 100 nm is also low at 10% [8,9]. Zhao et al. [10] selected cotton samples with different structures from a variety of materials. The results indicated that although the difference in gram weight (areal density) of cotton fabrics with different structures could be up to two times, the difference in filtration efficiency was not as significant, so the influence of gram weight and density on filtration efficiency was not straightforward. In the early days of the pandemic, Konda et al. [11] tested household fabrics such as cotton, silk, flannel, and chiffon with different yarn counts. It was found that under the same test conditions, the filter efficiency of single-layer high-density (600TPI) cotton fabric was about 70% higher than that of medium-density (80TPI) cotton fabric. The filter efficiency of double 80TPI cotton fabric is about 30% higher than that of single cotton fabric. The three fabrics have better filtering efficiency than N95 masks in blocking particles in the range of 10~300 nm. This important finding has not been thoroughly repeated and appeared to have had minimum impact on mask material choice in Western countries. Zangmeister et al. [12] tested the filtration efficiency of 32 kinds of household fabric materials (14 kinds of cotton, one kind of wool, nine kinds of synthetic, four kinds of polyester, and four kinds of polyester/cotton blend). Among the five samples with the highest filtration efficiency, three were pure cotton fabrics of high and medium weight, and the other two were medium-weight poly-cotton blended fabrics. In addition to measuring a single type of fabric, structures with two and multiple layers of fabric superimposed were also tested in the same or nearly the same combination according to Konda et al., but the wind speed and aerosol concentration were different, so the results of the two studies were inconsistent. Hao et al. [13] tested the filtration efficiency and flow resistance of 16 household materials (fabric and paper). The experimental results show that the filtration efficiency of paper and fabric generally increases with the increase of GSM value, and the fabric with a higher gram weight (GSM) has better quality and usability, which means that they can be used as a good candidate for homemade masks.

On the other hand, many highly efficient materials used in surgical mask materials are made up of fine nonwoven webs such as polypropylene (PP) melt-blown. This is partly due to the very fine fiber diameters in the PP melt-blown materials which yield much smaller pores in the fabrics. If the pore size of the braided fabric materials can be made sufficiently low, higher particle filtration efficiency can be expected. Whiley et al. [14] uses a viral aerosol in sterile water at a flow rate of 28.3 L/min for a homemade mask to compare the filtration efficiency (VFE) of 6 and 2.6 μm droplets to identify the following effective combinations: cotton mask plus a vacuum cleaner bag (VFE (6.0 μm) = 99.5%,VFE (2.6 μm) = 98.8%); cotton mask with a baby wiping cloth (VFE (6.0 μm) = 98.5%,VFE (2.6 μm) = 97.6%); and one layer of 100% linen plus one layer of poly film plus one layer of cheesecloth (VFE (6.0 μm) = 93.6%, VFE (2.6 μm) = 89.0%). The filtration efficiency of these combinations is comparable to that of regular medical and N95 masks.

We are taking a similar approach in this report, but in a more systemic way, combining various fabrics and nonwovens to test and compare the combined systems with their component materials. We further analyzed their porous structures and surface charge to gain further understanding on the filtration mechanism. Bar-On et al. put forward that the particle size of the SARS-CoV-2 is about 100 nm [15], which was thought to spread via droplets from coughing and sneezing. In terms of the aerosol particulate size range for the filtration testing, we are following the excellent Nature article by Liu et al. [16], who analyzed the occurrence and aerosol deposition of SARS-CoV-2 (COVID-19 virus) in the air of 30 monitoring sites in the initial pandemic-occurring places in the Wuhan public, including the People’s Hospital of Wuhan University. Their results showed that the live SARS-CoV-2 existed on particles in the diameter in the range of 250~500 nm. Conventional non-electret air filtration materials have a most penetrating particle size (MPPS) of 0.3 um [17,18].

Based on previous research, this paper will focus our filtration performance tests on the aerosol particle size range of 0.3 µm and 2.5 µm. We would like to explore the influential relationships between basic properties of different single-layer conventional fabrics and their aerosol filtration performances and to seek out favorable features of a fabric that give rise to relatively better filtration performance. Two different approaches have also been proposed in this study in an attempt to enhance filtration performances of conventional fabrics by homogeneous fabric combination and by heterogeneous fabric combination.

## 2. Materials and Experimental Methods

### 2.1. Materials

The collected materials are all woven fabric except Blend 3 (knitted fabric); the basic information is shown in Table 1. Common household fabrics, including three cotton fabrics (Cotton1, Cotton2, Cotton3), two wool fabrics (Wool1, Wool2), two polyester fabrics (Polyester1, Polyester2), two silk fabrics (Silk1, Silk2), and three blended fabrics (Blend1 (Cotton/PET 50/50), Blend2 (Wool/PET 50/50) and Blend3 (PET/PU/Wool 75/20/5)), were in total 12 representative conventional fabrics purchased in the market. (Qianchuan Department Store fabric market, Qingdao, SD, PRC.)The compositions, structures, thicknesses, fiber diameters, average pore sizes, and areal masses of these fabrics are listed in Table 1, which will be shown in the next section. In contrast with these common household fabrics, a polypropylene (PP) melt-blown nonwoven fabric was obtained from Shandong JOFO Nonwoven Co., Ltd., Weifang, SD, PRC.

### 2.2. Characterization

#### 2.2.1. Filtration Properties

The Automated Filter Media Test Rig AFC 131 (TOPAS GmbH, Dresden, GER) was used to evaluate the fractional filtration efficiencies of different fabrics and the pressure drop across the filter, with the use of NaCl aerosols having particle diameters ranging from 0.225 μm to 3.75 μm. As shown in Figure 1, the test rig consists essentially of the test channel with the test filter holder, the flow unit, the atomizer aerosol generator, and the optical particle measuring device. The fractional filtration efficiency and particle size distributions can be determined separately using an optical particle counter. Filtration performances against aerosols having particle diameters of 0.3 μm and 2.5 μm (PM2.5) were especially examined. The median particle diameter of NaCl aerosol was 0.3 μm, air flowrate was 32 L/min, and the test area, i.e., filter cross section, was 176 cm^2^, corresponding to a diameter of 150 mm. The filter efficiency calculation formula is:(1)$$\mathrm{FE}=\frac{C_{u-}C_{d}}{C_{u}}$$
where *C_u_* and *C_d_* are the mean particle concentrations per bin upstream and downstream, respectively.

#### 2.2.2. Fabric Texture and Microstructure

Use the optical microscope (OM) (WST-2KCH, Weishite Electronic Technology Co., LTD., Shenzhen, China) (SEM, Phenom Pro SEM, Hitzacker, Germany) to observe fabric textures and use the scanning electron microscope (SEM) to observe the microstructures in fabrics.

#### 2.2.3. Fabric Thickness

Use the YG141D digital fabric thickness meter to measure fabric thickness under 1 kPa pressure. The area of the presser foot is 500 mm^2^.

#### 2.2.4. Fiber Diameter

The software Nano measurer was used to measure the fiber diameter based on SEM images of fabrics. The average value was taken based on 100 fibers randomly selected.

#### 2.2.5. Average Pore Size

The pore size meter PSM 165 (TOPAS GmbH, Germany) was used to measure the average pore size of fabrics.

#### 2.2.6. Fabric Hand

Fabric hand refers to the tactile sensory perceptions generated when touching the fabric by hands or skin. The PhabrOmeter^®^ System (Nu Cybertek, US) was used to measure the hand feel performance of fabrics. Fabric samples were round with an area of 100 cm^2^. Six indicators can be obtained from a single measurement, including the drapability, wrinkle recovery, stretchability, resilience, softness, and smoothness.

#### 2.2.7. Water Vapor Transmission Rate (WVTR)

The water vapor transmission rates (WVTRs) (g/(m^2^ h)) for different fabrics were measured using YG601H-Ⅱ computerized fabric hygrometer.

#### 2.2.8. Air Permeability

Air Permeability Tester FX 3300-IV (TEXTEST AG, Schwerzenbach, CH) was used to measure the air permeabilities of fabrics. The test area was 20 cm^2^ and the test pressure level was 196 Pa.

## 3. Results and Discussion

### 3.1. Fabric Comfort and Filtration Performances

The feel of the fabric refers to the tactile effect of the fabric on hands and skin, which is an important part of evaluating the quality of the fabric and also the comfort level of people wearing fabric masks. Table 2 shows the fabric handle assessment of each experimental material. The larger the value, the better the performance. It can be seen that all the parameters of the fabric are similar to those of the melt-blown material, but the resilience of the fabric is about twice as high as that of the melt-blown material, which will have advantages in the design of masks and fitting accordingly.

Figure 2 shows the filtration efficiency and pressure drop of the above materials. When the aerosol particle size is 0.3 µm, the filtration efficiency of the single-layer fabric is significantly lower than that of the melt-blown material. Among the fabric groups, the filter efficiency of Wool 1, Blend 2, Cotton 1, and Cotton 3 are at a higher level in all fabrics. When the particle size is 2.5 µm, the filter efficiency of these three fabrics can reach 80%, so they can effectively resist PM 2.5.

In consideration on the performances, the melt-blown material is still the best choice for mask filtration application. However, the fabric made of natural fibers has the advantages of being comfortable to wear and more environmentally friendly [19]; with filtration efficiency improving towards the higher range of the aerosol particles, we can still see the fabric as having a certain application prospect. However, earlier reports on their high filtration efficiency could not be repeated.

Intuitively, we believe that the thicker the fabric, the higher the filtration efficiency may be; that is, the filtration efficiency of multi-layer fabric is higher than that of single-layer fabric. Therefore, several groups of representative samples with high single-layer filtration efficiency were selected for analysis. Although Cotton 1 showed the highest filtration efficiencies among three different cotton fabrics, it was not selected because of the overhigh pressure drop (>250 Pa) exceeding the maximum respiratory resistance in all criteria for the standard provisions of disposable medical masks, which can cause a feeling of suffocation, and so it is not suitable for facial mask material. The fabrics finally selected included Cotton 3, Wool 1, Polyester 2, Silk 1, Blend 1, and Blend 2. The thickness of the fabric sample with the same material and structure is uniform with the original material. In order to maintain the single variable of thickness, the thickness can only be changed by means of combination. The filtration efficiency and pressure drop of single-layer melt-blown fabrics were tested as the control group; the test results are shown in Table 3. Considering the influence of the increase of fabric layers on its air permeability, we made a comparison of the air permeability of these tissues.

Table 3 shows the comparison of filtration efficiency, pressure drop, air permeability, and quality factor Q of single- and double-layer fabrics at 0.3 µm and 2.5 µm, respectively. It is apparent that the stacked fabrics exhibited improved filtration efficiencies. For example, the filtration efficiency against 0.3 µm NaCl aerosol particles increased from 52.1% to 73.8% for double-layer Wool1, and it increased from 57.5% to 74.1% for double-layer Blend1; the filtration efficiency against 2.5 µm NaCl aerosol particles increased from 83.5% to 97.9% for double-layer Wool 1, and it increased from 85.8% to 94.9% for double-layer Blend1. Among these conventional stacked fabrics, the highest achievable filtration efficiency values against 2.5 µm NaCl aerosol particles were for Cotton 3 X2 (94.2%) (X2 indicates the two-layer fabric superposition), Wool 1 X2 (97.9%), and Blend 2 X2 (94.9%), even higher than that of the melt-blown fabric (93.4%). This demonstrates the potentials of using conventional fabrics as competent PM2.5 protective masks. On the other hand, the highest achievable filtration efficiency values against 0.3 µm NaCl aerosol particles were for Cotton 3 X2 (56.4%), Wool 1 (73.8%), and Blend 2 X2 (74.1%), which were still below that of the melt-blown fabric (83%). In other words, even if the pressure drop were disregarded, a conventional fabric is still considered unqualified as a facial mask for medical use; the standard YY0469-2011 Technical Requirements for Surgical Mask specifies that the bacterial filtration efficiency is ≥95% at the test flow rate of 28.3 L/min. Yet, as one of the critical measurements for filtration performance, the pressure drop cannot be ignored. In fact, a two-layer fabric achieved improved filtration efficiencies at the expense of deteriorated comfort due to an increase in pressure drop values. For instance, the pressure drop for Cotton 3 X2 (174 Pa) failed to meet the requirements of inspiratory resistance not exceeding 150 Pa and expiratory resistance not exceeding 120 Pa as specified in the Chinese Standard TAJ1001-2015 PM2.5 Protective Mask, although its PM2.5 filtration efficiency (94.2%) barely met the required filtration efficiency (95%) in this standard. Nevertheless, the pressure drop values of some other types of double-layer fabrics, e.g., Blend 2 X2 (12 Pa), were equal to that of the melt-blown fabric (12 Pa), which indicates the potential applicability of such double-layer fabrics in facial mask materials due to relatively smaller respiratory resistance and improved filtration efficiencies. It can be clearly seen from Table 3 that the increase of fabric layers will lead to the increase of pressure drop, accompanied by the decrease of air permeability, and the contrast between Cotton 3 and Silk 1 is particularly obvious. Therefore, it can be judged that the pressure drop and air permeability of the fabric have the same rule as that of the melt-blown material [20], as shown in the product section of Table 3. The pressure drop and air permeability are inversely proportional, and the two of the same pure fabric tends towards a fixed value, with the difference <1500.

Nevertheless, it is not enough to use only the thickness as an indicator to illustrate the barrier performance of fabric filters. Due to the complexity in fabric construction, multiple factors, and even a complex interplay of some of these factors, would exert their influences on the barrier performance of a fabric filter, such as linear densities of fibers and yarns, fiber length, fabric hairiness, fabric pattern, and stitch density. In addition, the effect of washing on the mask material filtration properties is analyzed in Appendix A (SI).

Since the wearing comfort of a facial mask also includes the aspect of moisture permeability, it is necessary to evaluate the water vapor transmission rates (WVTRs) of these different fabrics. Table 3 shows the moisture permeability of different single-layer fabrics. Conventional fabrics exhibited higher WVTRs than that of melt-blown fabric. A high WVTR is associated with better wearing comfort due to better transfer of moisture in the breath from mouth or nose into the environment, which has less influence on the service life of the mask. Therefore, the value of fabric is higher than that of melt-blown materials in terms of moisture permeability.

### 3.2. Influence of Surface Hairiness on Fabric Filtration Performance

As shown in Table 3, the filtration efficiency of double-layer Wool 1 and Blend 2 are higher than other double-layer fabrics. The scanning electron microscope (SEM) images of several fabrics are shown in Figure 3. Combined with the SEM image, it can be seen that when double-layer wool fabric is stacked in two layers, the wind resistance increases and it is not easy to be blown open by wind in the filtration process [21]. The increase of the gram weight and hair of the middle layer caused by the overlap of the two layers, which can better prevent the passage of small-particle-sized aerosol, thus increasing its filtration efficiency [22].

Although some fabrics and their combinations have higher filtration efficiency, the pressure drop increases with the increase of filtration efficiency, and the comfort level decreases. GB 19083-2010 Technical Requirements for Medical Protective Masks stipulates that when the gas flow rate is 85 L/min, the inhalation resistance of medical protective masks shall not exceed 343.2 Pa. In order to balance the relationship between protection and comfort, filtering quality factor (*Q*) is usually used for evaluation, which is an effective performance evaluation criterion that considers both protection ability and the comfort to wear [23]. The filtration quality factor (Q) can be calculated as [24]:(2)$$Q=\frac{-\ln(1-E)}{\Delta p}$$
where E is the filtration efficiency (0 ≤ E ≤ 100%) and Δp is the pressure drop (kPa). A higher filtration efficiency or a lower pressure drop is associated with a higher quality factor Q, or namely, better filtration performance.

Table 3 also shows the *Q* of the above-mentioned fabrics and melt-blown materials at 0.3 μm and 2.5 μm. The *Q* value of Cotton 3 was the lowest among all the samples although the filtration efficiency was very high. Fabric materials are generally lower than melt-blown materials, but we find that the quality factor of Blend 2 is much higher than melt-blown materials. This phenomenon may be due to its finer fibers, more and fluffy surface feathers, which provide more possibilities for blocking aerosol particles without increasing pressure drop. Through the SEM comparison of different fabrics, it can be seen that the common feature of Cotton 3, Wool 1, and Blend 2 is that there are fluffy and broken fibers on the surface. However, it can be seen from Table 1 that Cotton 3 has smaller pores, dense texture, and higher filtration efficiency, resulting in higher pressure drop and lower *Q* value. Wool 1 fabric is thicker and not compact, and large pores will be generated in the ventilation pipe during the filtration test, so the filtration effect is undesirable, and a low *Q* value will also be generated. The fiber diameter of Blend 2 fabric is at the medium level, resulted in a large aperture but high gram weight. Combining with SEM, it can be seen that the internal fibers of Blend 2 fabric are relatively close and the surface fibers are relatively fluffy, so the pressure drop of Blend 2 fabric is also in an ideal range while ensuring high filtration efficiency, thus achieving a high *Q* value.

Fabric surface hairiness increase the roughness and specific surface area of the fabric, in which the fiber loop space arrangement is staggered, and form a large number of three-dimensional systems of meandering channel to a certain extent. It can enhance the dispersion effect of fluid: the aerosols have more opportunity to collide with the single fiber and be caught [24], thus increasing the filtration efficiency. In addition, the fabric feather is fluffy and the air flow of human breathing can flow from the gap, so it will not produce a high pressure drop resulting in breathing difficulties (see Appendix A, SI).

### 3.3. Influence of Ambient Humidity Effect on Fabric Filtration Efficiency

Generally, fabrics are sensitive to ambient humidity—they are hygroscopic or deformable—so the influence of ambient humidity is indispensable when discussing the filtration performance of fabrics. The fabrics of the above six different materials were placed into an airtight circumstance with the relative humidity (RH) of 20%, 50%, and 80% for 5 h, and then taken out for the filtration efficiency test. As shown in Figure 4, with the RH increase, the filtration efficiency of all materials decreased at 0.3 um. However, at a particle size of 2.5 um, the filtration efficiency of the first three fabrics increased slightly, while the filtration efficiency of the last three fabrics showed a downward trend. Based on the filtration mechanism of non-electret materials, it can be seen that the decrease of filtration efficiency at 0.3 um is due to the weakening of the attraction of small-sized particles, which was caused by the loss of the fabrics’ surface static charge after the moisture absorption. At a particle size of 2.5 um, because Cotton 3 fibers are porous substances and there are lots of hydrophilic groups (−COOH and −OH) on the cellulose macromolecule chain, the hygroscopic property is good, as shown in the dynamic water contact angle (see Video S1, SI); Wool 1 fiber has a strong water-repellent outer layer, making it naturally water-repellent (see Figure 4g), but can absorb 35% of its own weight of water vapor—high moisture absorption; Silk 1 is mainly composed of silk fibroin and sericin, both of which contain a lot of polar groups, so it is easy to absorb water (see Video S2, SI). Most importantly, the hygroscopic expansion of natural fibers is remarkable, which results in coarser fibers and tighter fabric structure. In addition, it has a better blocking effect on the particles with larger particle sizes. Moreover, some particles can combine with the water molecules on the surface of the fabric materials to make the particle sizes larger, making it impossible for them to pass through the gaps of the fabric. Finally, the filtration efficiency increases at 2.5 um (see Figure 4b). Polyester 2, Blend 1, and Blend 2 fabrics contain more polyester fibers. Polyester exhibits hydrophobicity and poor hygroscopic properties; the water contact angle diagram of the fabrics was as shown in Figure 4f,h,i. Polyester molecules are linked by a covalent bond, which is non-ionizing and cannot transfer electrons. Meanwhile, there are few polar groups in polyester fiber; the charge dissipation is extremely difficult. These factors result in the ease to generate and accumulate electrical charges on the surface of polyester and high hydrophobicity. So that the humidity increases, polyester fabrics are not deformed but the surface charge is released to eliminate static electricity. Weakening of electrostatic effect reduces the adsorption of small-sized particles by fiber, so that the filtration efficiency is reduced.

At the same time, the coarser fibers and tighter fabric structure caused by hygroscopic expansion of the three natural fibers resulted in the increase of pressure drop (Figure 4c), so the quality factor decreased at 0.3 um, and was almost unchanged at 2.5 um. The structure of the last three kinds of fiber will not change due to the change of humidity, so the pressure drop basically does not change, and the filtration efficiency decreases while the quality factor decreases.

### 3.4. Influence of Triboelectric Effect on Silk Filtration Efficiency

The filter layer of medical masks and respiratory filters is usually combined with the melt-blown process and electret process, so that the melt-blown cloth has electrostatic properties and can effectively collect and block small particles, which can significantly improve its filtering efficiency without increasing any mass or compact structure [25,26].

It is well known that triboelectric effects are commonly used to demonstrate static electricity. Zhao et al. [10] rubbed the sample with a pair of latex gloves for 30 s and immediately recorded the filtration properties before and after treatment. The results showed that the filtration efficiency of the three cotton fabrics decreased or remained unchanged, while the filtration efficiency of the other samples increased. The decrease of filtration efficiency of cotton fabric may be due to the pore size expansion caused by friction or even the damage caused by wear on the sample. Therefore, mechanical damage, friction, or stretching can lead to a decrease in filtration efficiency, and these effects should be considered for cotton fabrics. On the other hand, all the other samples showed a moderate to high improvement in filtration efficiency when tested immediately after charging.

The use of latex and nitrile rubber friction fabric in improving filtration efficiency is the most significant. Latex and NBR (nitrile-butadiene rubber) are commonly used glove materials in our daily life, so it is relatively easy to use this material to charge fabric masks frequently (i.e., to rub the mask with a gloved hand before wearing the mask) [10]. Silk and glue stick friction is easy to produce static electricity, so choose silk 1 as a sample for experiment (See Appendix A (SI), the friction process of silk). The test results show that the pressure drop of Silk 1 is 7 Pa before and after friction, and the filter effect results are shown in Figure 5.

As can be seen from Figure 5, although the overall filtering efficiency of silk is low, the filtering efficiency of Silk 1 after friction is significantly greater than that before friction, indicating that friction can effectively improve the filtering efficiency of silk. When DEHS (Di-Ethyl-Hexyl-Sebacat) was used as an experimental aerosol, the particle size of MPPS was 0.08 μm [17], so the filtration efficiency showed a trend of first increasing and then decreasing. As shown in Figure 3d, the silk fabric is not tight enough; it will be blown apart in the filtration ventilation process, resulting in larger gaps, so the particle size increases, and its filtering efficiency decreases. At this time, electrostatic attraction is the main filtering mechanism, and electrostatic effect is the main mechanism to capture submicron particles. Therefore, it can adsorb aerosol particles with a smaller particle size and cannot adsorb aerosols with a larger particle size.

However, the static charge on the surface of the fabric will inevitably dissipate due to adsorption of water molecules in the air or discharge due to contact with other surfaces [27]. Zhao [10] evaluated potential attenuation in the natural environment (the sample was placed on a table top without any covering, and the temperature and humidity were roughly constant at 22 °C, 40% RH (Relative humidity). The experimental results show that the potential attenuation rate of polyester and silk is relatively fast, and the surface charge basically reaches the initial value in about 30 s. Therefore, the application of static electricity in fabric filtration needs further study.

### 3.5. The Superimposing Effect of the Silk and PLA Nano-Fabric on Filtration Efficiency

The combination of fabric and non-woven fabric has been effectively certified for the use in masks [28]. Figure 6 is a nanofiber-film super-protective mask purchased from the market. Figure 6b shows the outer, middle, and inner layer of the real nanofiber-film mask, and the corresponding SEM images of the surface and inner layer are shown in Figure 6c. It indicated that the mask is composed of two layers of micron fiber fabric and a nanofiber film in the middle. Figure 6a is the filtration performance test characterization of the nanofiber-film mask. Combined with Table 4, it can be seen that the filtration efficiency of the nanofiber mask at 0.3 μm is not high. Because the nanofiber film in the middle layer is thin and sparse, it does not completely cover the surface fabric, so although it cannot achieve a higher filtration efficiency, it is still higher than that of ordinary fabric. It can be inferred based on the above analysis that if the electrospinning nanofilm was completely laid on the micron fiber fabric, which was utilized as the collector, the composite fabric overall filtering efficiency of the mask will be improved to a large extent.

In order to further explore the effect of different fabric combinations on filtration efficiency, the filtration experiment of fabric and non-woven fabric combination was also done. Due to the low pressure drop and smooth surface of silk, it is easy to combine with non-woven fabric, and silk is a natural fiber, which is non-toxic to the human body and is degradable; therefore, we choose the combination of silk and non-woven fabric to complete this experiment. Considering the comfort of composite materials, the thickness, weight, filtration efficiency, and pressure drop of commercially available polylactic acid (PLA) non-woven fabric, polypropylene (PP) non-woven fabric, and PLA electrostatic spinning non-woven fabric were compared with PP melt-blow non-woven fabric in the middle layer of masks [29,30]. Finally, the combination of PLA electrostatic spinning non-woven fabric and silk was selected. As shown in Appendix A, the gram weight, thickness, and pressure drop of PLA electrostatic spinning non-woven cloth are much smaller than other materials. When combined with silk, the weight and resistance of the composite material will not be increased. As a degradable material with high filtration efficiency [31,32], PLA electrostatic spinning non-woven cloth is a better choice for combination masks.

Two combinations of PLA directly spun on silk (PLA/silk) and the spun PLA covered on silk (PLA + silk) were tested, respectively. The spinning conditions were the same (see SI, the electrospinning process of PLA fibers) and the time was 30 min. The results were compared with the melt-blown out layer inside the disposable medical mask, as shown in Figure 7 (The single-layer PLA membrane used the rigid mesh non-woven fabric as the base, and its filtration efficiency and pressure drop were negligible).

It can be seen from the figure that the filtration efficiency of monolayer PLA and its combination with silk is between 85% and 100%, basically the same as that of monolayer melt-blown. Although the filter efficiency of monolayer PLA, PLA spun on silk, and covered on silk are not much different, the filter efficiency of the latter two are basically lower than the former, especially in the 0.225–0.525 μm particle size. According to the aerosol filtration mechanism, the preliminary guess is that the combination of silk and electrospinning membrane results in partial electrostatic neutralization on the surface of both, and aerosols with a small particle size cannot be adsorbed by electrostatic force, thus reducing the filtration efficiency.

In order to verify the above surmise, a Ningbo Textile Factory electrostatic attenuator was used to conduct the following electrostatic attenuator tests: monolayer electrospinning membrane for 30 min and 35 min, and the combination of the two electrospinning membrane with silk, respectively. Compare the results after 30 s of electrostatic attenuation. Due to the use of high-voltage discharge during the test, the initial static electricity of the sample cannot be displayed. Therefore, voltage comparison is not made, and only electrostatic attenuation before and after the electrospinning membrane is combined with silk is considered. The results are shown in Table 5.

As can be seen from the above table, the voltage of the PLA membrane alone hardly attenuates after 30 s, and the attenuation rate is 1–5%. The decay rate of pure silk fabric is higher, the voltage stored after 30 s is very small, and the decay rate is 96%. Whether PLA is spun directly on silk or covered on silk, its voltage attenuation is large, and its attenuation rate is between 58% and 70%. It can be seen that the electrospinning membrane can store electric charge well and adsorb aerosols with a small particle size, thus achieving higher filtration efficiency in the small particle size range. Although silk can obtain a higher voltage through discharge, it will escape quickly in a short time, so its main filtration mechanism is not electrostatic adsorption. When PLA is combined with silk, the electrostatic attenuation rate will increase by more than 10 times and the surface charge will be reduced. Therefore, combining electrostatic spinning with fabric will have the risk of reducing filtering efficiency, as shown in Figure 7, which is obvious in the range of small particle size. Although the combination of the two reduces the filter efficiency, silk has little influence on the PLA membrane as long as it has a high filter efficiency. Figure 8 shows the filter efficiency comparison of PLA and its combination (the pressure drop of pure PLA is 18 Pa, and the pressure drop of silk combination is 20 Pa) when spinning for 40 min. At this time, whether PLA is combined with silk or not has little impact on the filter efficiency and pressure drop, and even increases the filter efficiency (0.225~0.525 μm). Therefore, PLA electrostatic spinning membrane combined with silk can be considered as the filter.

## 4. Conclusions

Having tested several types of fabric materials as mask candidates in their filtration efficiency of particulate matter in the size range of 0.225 μm to 3.750 μm, we found that: (1) Fabrics with a quality factor higher than that of the common disposable medical masks can be found from mixed wool fabrics common in daily use. Fluffy fabric structure and fine hairs on the surface showed superior filtration performance in blocking aerosol particles; (2) The triboelectric effect can increase the filtration efficiency of silk. Because silk fabrics are typically woven with bigger micropores, electrostatic attraction is the main filtration mechanism, so aerosol particles with a small particle size could still be absorbed; (3) The comfort factor in moisture permeability for the above fabric is higher than that of the melt-blown material. When the fabric is used as a protective mask, the water vapor generated by respiration can be transmitted outward through the fabric, and some fabrics have higher filtration efficiency, meeting the requirements of PM2.5 protective masks; (4) The combination of woven fabric and electrospun nanofiber membranes could lead to the reduction of static charge and the reduction of filtration properties on nano-scale particles until the nanofiber layer is sufficiently thick. In conclusion, the use of fabric masks can provide significant protection against aerosol particle transmission within a certain range, and the comprehensive comfort of fabric and fit to facial contours can be higher than that of the current synthetic polypropylene based masks.

Most of the current masks are consumer products, not medical supplies, and therefore may not meet the medical/hospital standards and should not be used for treatment in any hospital. Future studies should include the crucial product design of fabric masks so they can be more comfortable and fit well during daily use. The effect of repeated use (laundering) of fabric products should also be further evaluated.

## Figures and Tables

**Figure 1 nanomaterials-13-00378-f001:**
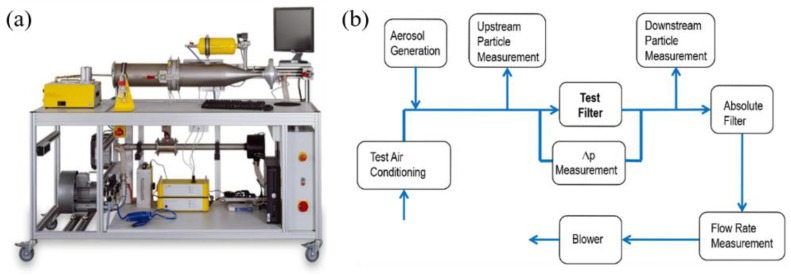
(**a**) The experimental device and (**b**) the schematic diagram of the filter media test rig.

**Figure 2 nanomaterials-13-00378-f002:**
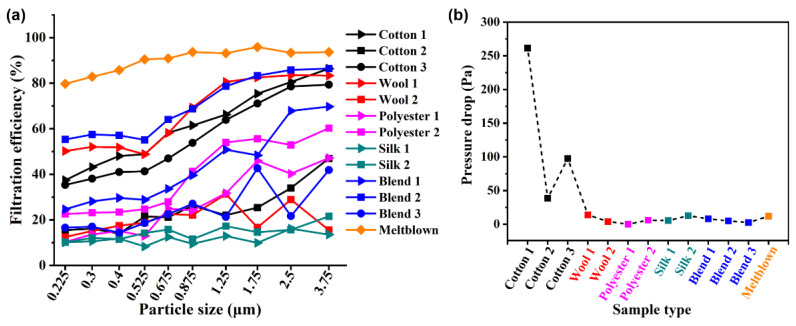
(**a**) The filtration efficiency in the range from 0.225 μm to 3.75 μm and (**b**) the pressure drop for the 13 samples.

**Figure 3 nanomaterials-13-00378-f003:**
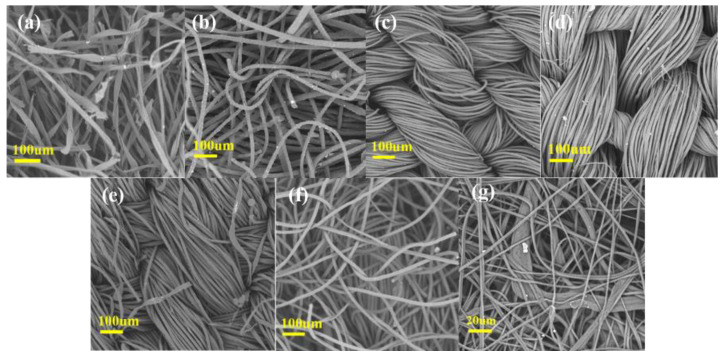
SEM images of (**a**) Cotton 3, (**b**) Wool 1, (**c**) Polyester 2, (**d**) Silk 1, (**e**) Blend 1, (**f**) Blend 2, and (**g**) Melt-blown fabrics.

**Figure 4 nanomaterials-13-00378-f004:**
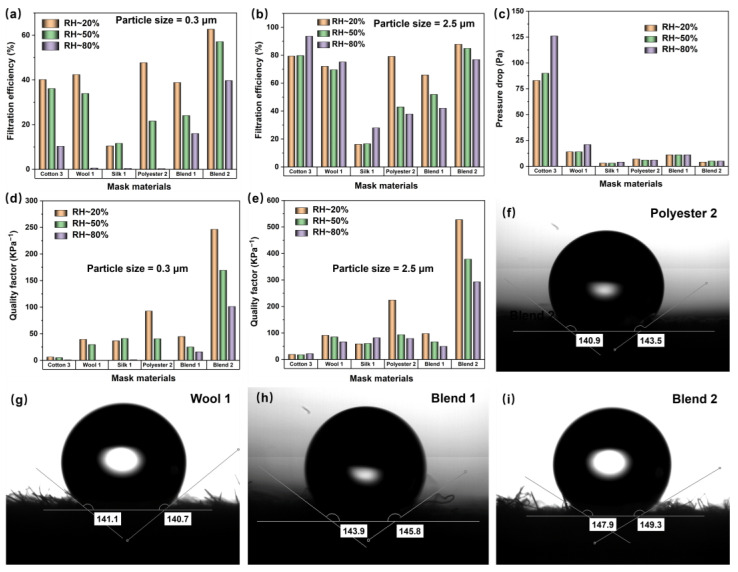
(**a**,**b**) Filtration efficiency, (**c**) pressure drop, and (**d**,**e**) quality factor of six fabrics at particle size of 0.3 um and 2.5 μm. Water contact angle of (**f**) Polyester 2, (**g**) Wool 1, (**h**) Blend 1, and (**i**) Blend 2 fabrics.

**Figure 5 nanomaterials-13-00378-f005:**
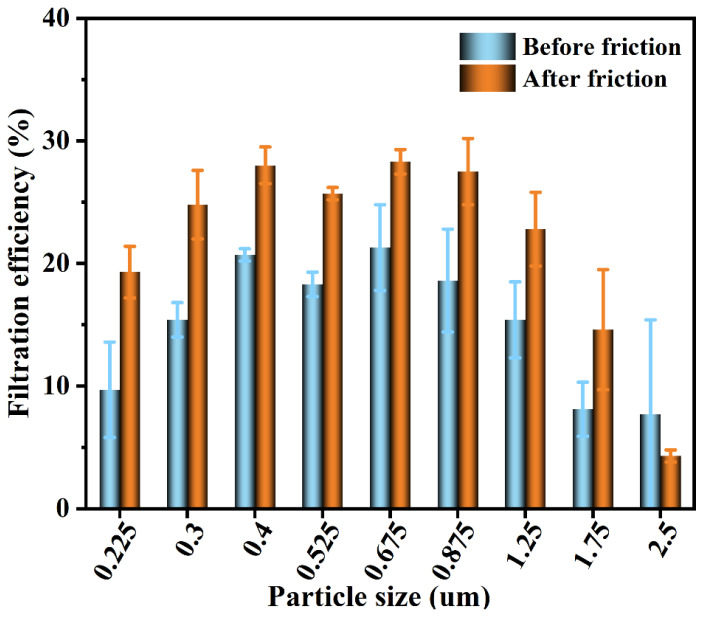
Filtration efficiency of silk before and after friction under DEHS aerosol.

**Figure 6 nanomaterials-13-00378-f006:**
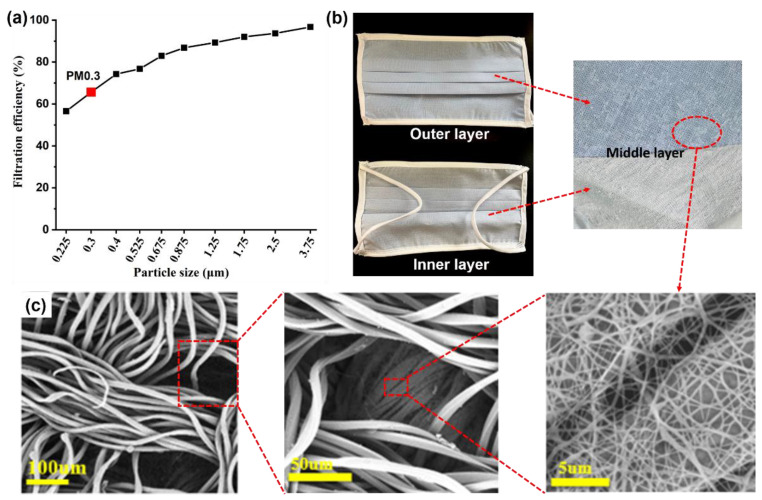
(**a**) The filtration efficiency curve, (**b**) photographs, and (**c**) SEM images of nanofiber film mask.

**Figure 7 nanomaterials-13-00378-f007:**
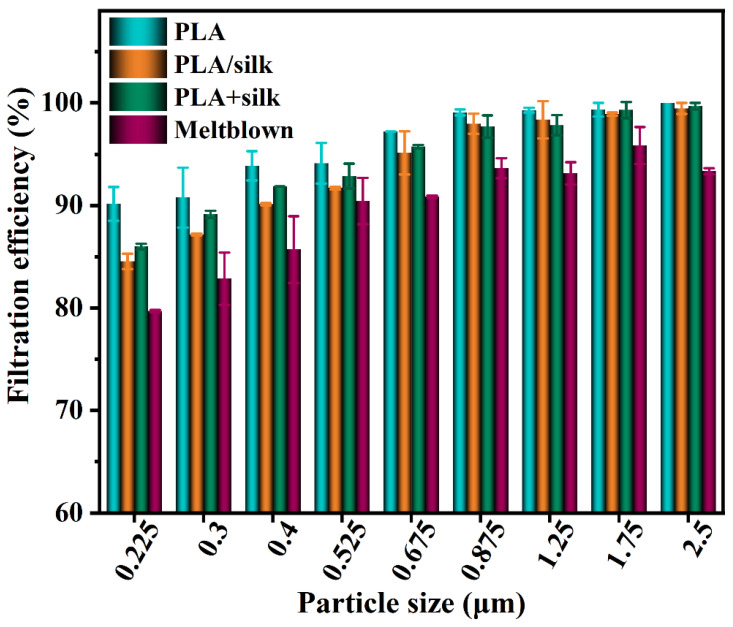
Filtration efficiency of electrospinning film (PLA), PLA/silk, PLA+silk, and melt-blown samples.

**Figure 8 nanomaterials-13-00378-f008:**
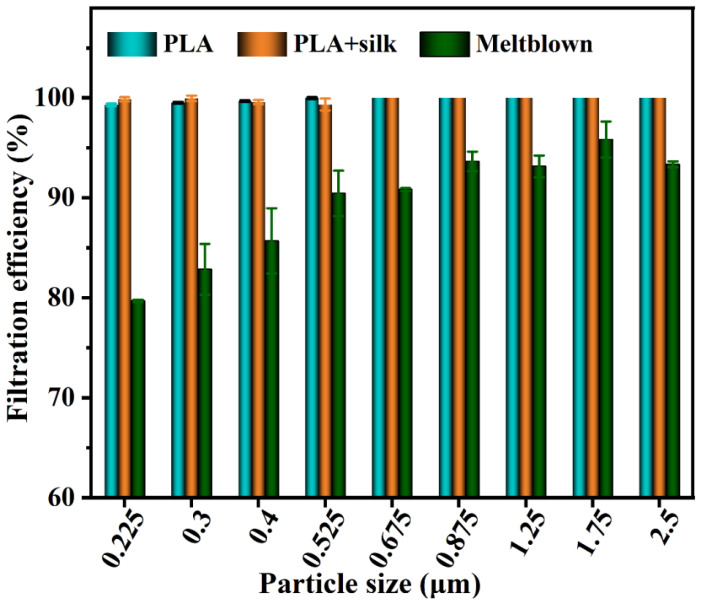
The filter efficiency of PLA spun for 40 min, PLA+silk, and melt-blown samples.

**Table 1 nanomaterials-13-00378-t001:** The basic structure parameters of these fabrics.

Fabric Code	Composition	Weaving Pattern	Thickness (mm)	Fiber Diameter (µm)	Average Aperture (µm)	Gram Weight (g/m^2^)
Cotton 1	100% Cotton	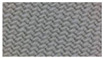	0.422	14.16	7.19	253.00
Cotton 2	100% Cotton	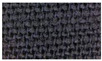	0.380	12.31	24.28	207.58
Cotton 3	100% Cotton	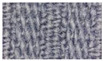	1.200	16.80	13.15	366.50
Wool 1	100% Wool	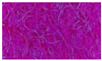	1.200	21.04	36.61	382.08
Wool 2	100% Wool	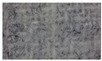	0.891	17.31	74.23	259.92
Polyester 1	100% Polyester	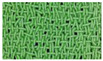	0.658	13.50	136.14	306.25
Polyester 2	100% Polyester		0.387	9.61	68.38	178.83
Silk 1	100% Silk	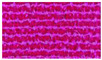	0.443	12.13	77.93	182.00
Silk 2	100% Silk	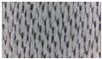	0.256	10.93	37.09	129.25
Blend 1	30% Cotton/70% Polyester	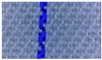	0.304	11.54	33.57	136.17
Blend 2	20% Wool/80% Polyester	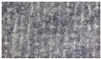	1.578	16.10	46.27	490.08
Blend 3	5% Wool/20% Spandex /75% Polyester	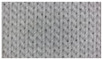	0.613	12.75	80.20	218.75
Melt-blown	100% Polypropylene	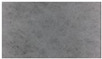	0.166	2.42	16.59	21.33

**Table 2 nanomaterials-13-00378-t002:** Fabric handle assessment parameters of samples.

Sample	Drape	Wrinkle Recover Rate (%)	Stretch (Trial)	Resilience Score	Softness Score	Smoothness Score
Cotton 1	32.18	64.28	0.09	53.78	45.41	57.09
Cotton 2	29.42	66.26	0.19	49.8	48.01	48.63
Cotton 3	12.69	69.58	0.24	40.69	63.32	48.49
Wool 1	19.43	80.79	0.43	51.06	59.8	48.05
Wool 2	22.33	75.02	0.5	57.85	56.44	52.54
Polyester 1	13.13	94.13	0.26	42.6	60.85	44.47
Polyester 2	16.27	95.94	0.3	32.43	63.81	46.79
Silk 1	36.67	79.47	0.55	67.45	69.03	68.14
Silk 2	14.55	81.4	0.33	30.5	64.13	45.69
Blend1	8.13	96.8	0.74	22.1	75.72	59.23
Blend2	21.84	79.57	0.42	54.56	57.94	47.69
Blend3	14.4	89.59	0.18	29.05	60.65	49.46
Melt-blown	11.94	78.65	0.32	28.78	59.84	55.48

**Table 3 nanomaterials-13-00378-t003:** Filtration and air permeability performances for single and double layers of different fabrics.

Sample Type	Filtration Efficiency (%)		Pressure Drop (Pa)	Air Permeability (mm/s)	Product	*Q* (kPa^−1^)		Moisture Permeability (g/m^2^·h)
	0.3 um	2.5 um				0.3 μm	2.5 μm	
Cotton3 X1	38.05	78.55	97.5	62.0	6045	4.9	15.8	206.1
Cotton3 X2	56.4	94.2	174	34.5	6003	4.8	16.4	-
Wool1 X1	52.1	83.5	14	340	4760	52.6	128.8	208.7
Wool1 X2	73.8	97.9	33	164	5412	40.6	116.9	-
Polyester2 X1	23.15	52.9	6	620	3720	43.9	125.5	235.3
Polyester2 X2	31	69.7	19	264.7	5029.3	19.5	62.8	-
Silk1 X1	10.6	16.15	5.5	1370	7535	20.4	32.0	210.9
Silk1 X2	17.6	16.6	14.5	462.3	6703.35	13.4	12.5	-
Blend1 X1	28.15	67.85	8	391	3128	41.3	141.8	175.6
Blend1 X2	49	85	29	201.7	5849.3	23.2	65.4	-
Blend2 X1	57.5	85.8	5	618.7	3093.5	171.1	390.1	264.5
Blend2 X2	74.1	94.9	12	389	4668	112.6	248.2	-
Melt-blown	82.85	93.35	12	520.7	6248.4	146.9	225.9	162.5

**Table 4 nanomaterials-13-00378-t004:** Basic parameters of the mask.

Fiber Diameter (um)		Efficiency of Filtration (%)		Pressure Drop (Pa)	Gram Weight (g/m^2^)	Thickness (mm)
Outer Layer	Middle Layer	0.3 um	2.5 um			
12.52	0.15	65.7	93.7	30	99	0.31

**Table 5 nanomaterials-13-00378-t005:** The electrostatic attenuation data of samples.

Sample	Initial Voltage (V)	Voltage after Attenuation (V)	Decay Rate (%)
30 min PLA	229	217	5
30 min PLA/silk	1074	323	70
30 min PLA+silk	1336	558	58
35 min PLA	161	159	1
35 min PLA/silk	1246	500	60
35 min PLA+silk	1107	358	68
silk	864	33	96

## Data Availability

Not applicable.

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
