# Peer review of "Evaluation of Mask Performances in Filtration and Comfort in Fabric Combinations"

_nanomaterials, 2023, doi:10.3390/nano13030378_

Round 1

Reviewer 1 Report

The manuscript by Wang et al. describes an investigation into the capabilities of different fabric types and combinations of fabrics for filtering out microscopic aerosol particles from air streams.  In my opinion, the manuscript is fairly well written and the work reported is interesting.  Consequently, I would be happy to recommend publication, subject to addressing a few comments:

L113-114: Since the desire to remove airborne SARS-CoV-2 virus features strongly in the work, it would also be useful to note that the virus itself is around 100 nm in diameter, e.g. see:

Bar-On et al. eLife 2020;9:e57309.
DOI: https://doi.org/10.7554/eLife.57309

L134-135: I could not find the Supplementary Information.  Nevertheless, I suggest that basic information about the fabric samples used is sufficiently important to deserve a place in the main text.

L169: How were the aerosol particles generated and what method was used to characterise their diameters, please?

Was the characterisation really accurate to 5 nm?  Can the authors comment, please?

Table 1: In order to appreciate whether the differences between these materials were significant, what were the uncertainty limits, please?  

Do I understand that three samples were used for each measurements?  If so, can the authors indicate the ranges for the measurements in the table, please.

It would be very useful to know what were the fabric constructions of the different materials.  E.g. were they knitted, woven or 'non-woven' - and if woven, what sort of weaving patterns.

Fabric thickness may also be an important parameter to include in this table.

Table 2: I suggest it would be useful to indicate in the caption that the data referred to single or double layers, e.g: 'Table 2: Filtration and air permeability performances for single and double layers of different fabrics'.

L234-235: The authors note that 'a two-layer fabric achieved improved filtration efficiencies'.  Is that because two layers was thicker than one, giving an element of 'depth filtration'?

I suggest the authors should include thickness as a key parameter.

L254: The authors state that 'Conventional fabrics exhibited higher WVTRs than that of Meltblown fabric.'  Is that because the 'conventional fabrics' were more hydrophilic than the melt-blown material, which I presume to be some kind of thermoplastic?  The authors should indicate the chemical composition of the melt-blown material, please?

Eqn. 1:  This formula does not work if E is a %age, as in Tables 2, 3 and Fig. 3.  I suggest the authors should re-write this formula with E and a %age - or change the way it is presented elsewhere in the manuscript, to be consistent.

It would also be useful to explain how the filtration efficiency was calculated please.

L303: I suggest it would be useful for the authors to discuss how the test aerosols passed through the filters?  Was it possible that liquid condensed on the fabric, then gradually flowed through, driven by the air pressure, then formed new droplets on the 'downstream' side of the filter?  Can the authors comment please?

L335: What are DEHS and MPPS?  Please define these abbreviations in the text.

There were also a numbe of typographical or grammatical errors:

L48-58: I think the authors mean 'systematic' rather than 'systemic'.

L98 - 100: It appears the name of the author being cited is Whiley.  (H is the initial of his/her given name.)  Linked to that, it would be better just to state 6 and 2.6 µm droplets, rather than 'Whileyh (6 µm)' and 'Whileyh (2.6 µm)'.

L303: '...and be caught...'

L315-316: ...as the initial charge of the sample...  It looks like something is missing from this sentence.  Please check.

L364-365:  The phrase '...nanofilm was completely lied on the micron fiber fabric which was utilize as the collector...' contains two mistakes.  I suggest: '...nanofilm was completely laid on the micron fiber fabric, which was utilized as the collector....'

L375: The expression '...which is easy to non-toxic and degradable...' is not clear.  Do the authors mean '...which is easy to detoxify and degrade...'  The authors should check, please.

L441: '...wool fabrics common in daily use...'

L454: '...facial contours can be higher than...'

Author Response

Please see the attrchment.

Reviewer 2 Report

In this work fabrics, nonwovens and their combinations made of cotton, silk, wool, and synthetic fibers are tested toward filtration efficiency for aerosol particles with diameters ranging from 0.225 μm to 3.750 μm according under industrial standard testing conditions. Evaluation of the mask performance is crucial for ensuring protection of  people from bacteria and viruses, such as COVID-19 virus.  However, some points need to be further addressed:

- what is the justification for the „Nanomaterials” journal selection?  There are nanomaterials (dimensuions < 100 nm) applied in this work;

- KEY COMMENT:  Please provide more characteristics of the materials used (cotton, wool, polyester, silk, cotton-polyester blended fabrics and other fabrics) and present structure-property relationships;

- how the conditions for testing the filtration performance were selected?

- what is the filtration quality of cotton and wool masks after washing?

Reviewer 3 Report

Although the paper is moderately interesting, it should be improved according to the following lines:

1- A chart should be presented to show the relation between surface hairiness and fabric filtration performance.

2- What is the relation between the filtration properties and the thickness of mask? More discussion should be added.

3- What is the relation between the chemical structure of fibers used and the filtration properties? More discussion should be added.

4- What is the effect of relative humidity of masks environment on the filtration properties? It should be measure and the relations should be discussed.

Round 2

Reviewer 2 Report

Authors provided proper explanations  to the reviewer’s comments, and the revised manuscript can be published.

Reviewer 3 Report

It is acceptable now.